# Deep Joint Task Learning for Generic Object Extraction

**Xiaolong Wang**[1,4]**, Liliang Zhang**[1]**, Liang Lin**[1,3]***, Zhujin Liang**[1]**, Wangmeng Zuo**[2]
[1]Sun Yat-sen University, Guangzhou 510006, China
[2]School of Computer Science and Technology, Harbin Institute of Technology, China
[3]SYSU-CMU Shunde International Joint Research Institute, Shunde, China
[4]The Robotics Institute, Carnegie Mellon University, Pittsburgh, U.S.
xlwang@cmu.edu, linliang@ieee.org

## Abstract

This paper investigates how to extract objects-of-interest without relying on hand-craft features and sliding windows approaches, that aims to jointly solve two sub-tasks: (i) rapidly localizing salient objects from images, and (ii) accurately segmenting the objects based on the localizations. We present a general joint task learning framework, in which each task (either object localization or object segmentation) is tackled via a multi-layer convolutional neural network, and the two networks work collaboratively to boost performance. In particular, we propose to incorporate latent variables bridging the two networks in a joint optimization manner. The first network directly predicts the positions and scales of salient objects from raw images, and the latent variables adjust the object localizations to feed the second network that produces pixelwise object masks. An EM-type method is presented for the optimization, iterating with two steps: (i) by using the two networks, it estimates the latent variables by employing an MCMC-based sampling method; (ii) it optimizes the parameters of the two networks unitedly via back propagation, with the fixed latent variables. Extensive experiments suggest that our framework significantly outperforms other state-of-the-art approaches in both accuracy and efficiency (e.g. 1000 times faster than competing approaches).

## 1 Introduction

One typical vision problem usually comprises several subproblems, which tend to be tackled jointly to achieve superior capability. In this paper, we focus on a general joint task learning framework based on deep neural networks, and demonstrate its effectiveness and efficiency on generic (i.e., category-independent) object extraction.

Generally speaking, two sequential subtasks are comprised in object extraction: rapidly localizing the objects-of-interest from images and further generating segmentation masks based on the localizations. Despite acknowledged progresses, previous approaches often tackle these two tasks independently, and most of them applied sliding windows over all image locations and scales [17, 22], which could limit their performances. Recently, several works [33, 18, 5] utilized the interdependencies of object localization and segmentation, and showed promising results. For example, Yang et al. [33] introduced a joint framework for object segmentations, in which the segmentation benefits from the object detectors and the object detections are in consistent with the underlying segmentation of the

---

*Corresponding author is Liang Lin. This work was supported by the National Natural Science Foundation of China (no.61173082), the Hi-Tech Research and Development Program of China (no.2012AA011504), Guangdong Science and Technology Program (no. 2012B031500006), Special Project on Integration of Industry, Educationand Research of Guangdong (no.2012B091000101), and Fundamental Research Funds for the Central Universities (no.14lgjc11).

image. However, these methods still rely on the exhaustively searching to localize objects. On the other hand, deep learning methods have achieved superior capabilities in classification [21, 19, 23] and representation learning [4], and they also demonstrate good potentials on several complex vision tasks [29, 30, 20, 25]. Motivated by these works, we build a deep learning architecture to jointly solve the two subtasks in object extraction, in which each task (either object localization or object segmentation) is tackled by a multi-layer convolutional neural network. Specifically, the first network (i.e., localization network) directly predicts the positions and scales of salient objects from raw images, upon which the second network (i.e., segmentation network) generates the pixelwise object masks.

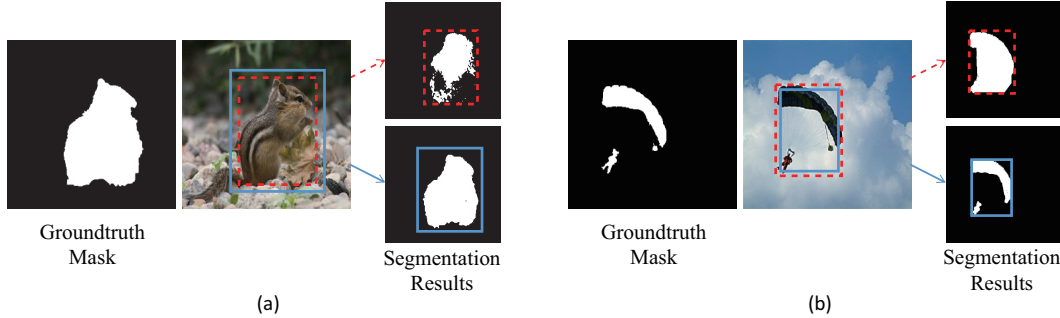

Groundtruth Mask      Segmentation Results      (a)      Groundtruth Mask      Segmentation Results      (b)

Figure 1: Motivation of introducing latent variables in object extraction. Treating predicted object localizations (the dashed red boxes) as the inputs for segmentation may lead to unsatisfactory segmentation results, and we can make improvements by enlarging or shrinking the localizations (the solid blue boxes) with the latent variables. Two examples are shown in (a) and (b), respectively.

Rather than being simply stacked up, the two networks are collaboratively integrated with latent variables to boost performance. In general, the two networks optimized for different tasks might have inconsistent interests. For example, the object localizations predicted by the first network probably indicate incomplete object (foreground) regions or include a lot of backgrounds, which may lead to unsatisfactory pixelwise segmentation. This observation is well illustrated in Fig. 1, where we can obtain better segmentation results through enlarging or shrinking the input object localizations (denoted by the bounding boxes). To overcome this problem, we propose to incorporate the latent variables between the two networks explicitly indicating the adjustments of object localizations, and jointly optimize them with learning the parameters of networks. It is worth mentioning that our framework can be generally extended to other applications of joint tasks in similar ways. For concise description, we use the term "segmentation reference" to represent the predicted object localization plus the adjustment in the following.

For the framework training, we present an EM-type algorithm, which alternately estimates the latent variables and learns the network parameters. The latent variables are treated as intermediate auxiliary during training: we search for the optimal segmentation reference, and back tune the two networks accordingly. The latent variable estimation is, however, non-trivial in this work, as it is intractable to analytically model the distribution of segmentation reference. To avoid exhaustively enumeration, we design a data-driven MCMC method to effectively sample the latent variables, inspired by [24, 31]. In sum, we conduct the training algorithm iterating with two steps: (i) Fixing the network parameters, we estimate the latent variables and determine the optimal segmentation reference under the sampling method. (ii) Fixing the segmentation reference, the segmentation network can be tuned according to the pixelwise segmentation errors, while the localization network tuned by taking the adjustment of object localizations into account.

## 2  Related Works

Extracting pixelwise objects-of-interest from an image, our work is related to the salient region/object detections [26, 9, 10, 32]. These methods mainly focused on feature engineering and graph-based segmentation. For example, Cheng et al. [9] proposed a regional contrast based saliency extraction algorithm and further segmented objects by applying an iterative version of GrabCut. Some approaches [22, 27] trained object appearance models and utilized spatial or geometric priors to address this task. Kuettel et al. [22] proposed to transfer segmentation masks from training data

into testing images by searching and matching visually similar objects within the sliding windows. Other related approaches [28, 7] simultaneously processed a batch of images for object discovery and co-segmentation, but they often required category information as priors.

Recently resurgent deep learning methods have also been applied in object detection and image segmentation [30, 14, 29, 20, 11, 16, 2, 25]. Among these works, Sermanet et al. [29] detected objects by training category-level convolutional neural networks. Ouyang et al. [25] proposed to combine multiple components (e.g., feature extraction, occlusion handling, and classification) within a deep architecture for human detection. Huang et al. [20] presented the multiscale recursive neural networks for robust image segmentation. These mentioned methods generally achieved impressive performances, but they usually rely on sliding detect windows over scales and positions of testing images. Very recently, Erhan et al. [14] adopted neural networks to recognize object categories while predicting potential object localizations without exhaustive enumeration. This work inspired us to design the first network to localize objects. To the best of our knowledge, our framework is original to make the different tasks collaboratively optimized by introducing latent variables together with network parameter learning.

## 3  Deep Model

In this section, we introduce a joint deep model for object extraction(i.e., extracting the segmentation mask for a salient object in the image). Our model is presented as comprising two convolutional neural networks: localization network and segmentation network, as shown in Fig. 2. Given an image as input, our first network can generate a 4-digit output, which specifies the salient object bounding box(i.e. the object localization). With the localization result, our segmentation network can extract a $m \times m$ binary mask for segmentation in its last layer. Both of these networks are stacked up by convolutional layers, max-pooling operators and full connection layers. In the following, we introduce the detailed definitions for these two networks.

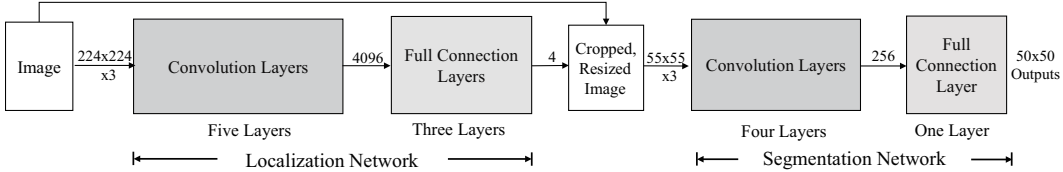

Figure 2: The architecture of our joint deep model. It is stacked up by two convolutional neural networks: localization network and segmentation network. Given an image, we can generate its object bounding box and further extract its segmentation mask accordingly.

**Localization Network.** The architecture of the localization network contains eight layers: five convolutional layers and three full connection layers. For the parameters setting of the first seven layers, we refer to the network used by Krizhevsky et al. [21]. It takes an image with size $224 \times 224 \times 3$ as input, where each dimension represents height, width and channel numbers. The eighth layer of the network contains 4 output neurons, indicating the coordinates $(x_1, y_1, x_2, y_2)$ of a salient object bounding box. Note that the coordinates are normalized w.r.t. image dimensions into the range of $0 \sim 224$.

**Segmentation Network.** Our segmentation network is a five-layer neural network with four convolutional layers and one full connection layer. To simplify the description, we denote $C(k, h \times w \times c)$ as a convolutional layer, where $k$ represents kernel numbers, and $h, w, c$ represent the height, width and channel numbers for each kernel. We also denote $FC$ as a full connection layer, $RN$ as a response normalization layer, and $MP$ as a max-pooling layer. The size of max-pooling operator is set as $3 \times 3$ and the stride for pooling is 2. Then the network architecture can be described as: $C(256, 5 \times 5 \times 3) - RN - MP - C(384, 3 \times 3 \times 256) - C(384, 3 \times 3 \times 384) - C(256, 3 \times 3 \times 384) - MP - FC$. Taking an image with size $55 \times 55 \times 3$ as input, the segmentation network generates a binary mask with size $50 \times 50$ as the output from its full connection layer.

We then introduce the inference process as object extraction. Formally, we define the parameters of the localization network and segmentation network as $\omega^l$ and $\omega^s$, respectively. Given an input image $I_i$, we first resize it to $224 \times 224 \times 3$ as the input for the localization network. Then the output

of this network via forward propagation is represented as $F_{\omega^l}(I_i)$, which indicates a 4-dimension vector $b_i$ for the salient object bounding box. We crop the image data for salient object according to $b_i$, and resize it to $55 \times 55 \times 3$ as the input for the segmentation network. By performing forward propagation, the output for segmentation network is represented as $F_{\omega^s}(I_i, b_i)$, which is a vector with $50 \times 50 = 2500$ dimensions, indicating the binary segmentation result for object extraction.

## 4 Learning Algorithm

We propose a joint deep learning approach to estimate the parameters of two networks. As the object bounding boxes indicated by groundtruth object mask might not provide the best references for segmentation, we embed this domain-specific prior as latent variables in learning. We adjust the object bounding boxes via the latent variables to mine optimal segmentation references for training. For optimization, an EM-type algorithm is proposed to iteratively estimate the latent variables and the model parameters.

### 4.1 Optimization Formulation

Suppose a set of $N$ training images are $I = \{I_1, ..., I_N\}$, the segmentation masks for the salient objects in them are $Y = \{Y_1, ..., Y_N\}$. For each $Y_i$, we use $Y_i^j$ to represent its $j$th pixel, and $Y_i^j = 1$ indicates the foreground, while $Y_i^j = 0$ the background. According to the given object masks $Y$, we can obtain the object bounding boxes around them tightly as $L = \{L_1, ..., L_N\}$, where $L_i$ is a 4-dimensional vector representing the coordinates $(x_1, y_1, x_2, y_2)$. For each sample, we introduce a latent variable $\Delta L_i$ as the adjustment for $L_i$. We name the adjusted bounding box as segmentation reference, which is represented as $\widetilde{L}_i = L_i + \Delta L_i$. The learning objective is defined as maximizing the probability:

$$P(\omega^l, \omega^s, \widetilde{L}|Y, I) = P(\omega^l, \widetilde{L}|Y, I)P(\omega^s, \widetilde{L}|Y, I), \tag{1}$$

where we need to jointly optimize the model parameters $\omega^l, \omega^s$, and the segmentation references $\widetilde{L} = \{\widetilde{L}_1, ..., \widetilde{L}_N\}$ indicated by the latent variables. The probability $P(\omega^l, \omega^s, \widetilde{L}|Y, I)$ can be decomposed into the probability for localization network $P(\omega^l, \widetilde{L}|Y, I)$ and the one for segmentation network $P(\omega^s, \widetilde{L}|Y, I)$.

For the localization network, we optimize the model parameters by minimizing the Euclidean distance between the output $F_{\omega^l}(I_i)$ and the segmentation reference $\widetilde{L}_i = L_i + \Delta L_i$. Thus the probability for $\omega^l$ and $\widetilde{L}$ can be represented as,

$$P(\omega^l, \widetilde{L}|Y, I) = \frac{1}{Z} \exp(-\sum_{i=1}^{N} ||F_{\omega^l}(I_i) - L_i - \Delta L_i||_2^2), \tag{2}$$

where $Z$ is a normalization term.

For the segmentation network, we specify each neuron of the last layer as a binary classification output. Then the parameters $\omega^s$ are estimated via logistic regression,

$$P(\omega^s, \widetilde{L}|Y, I) = \prod_{i=1}^{N} (\prod_{\{j|Y_i^j=1\}} F_{\omega^s}^j(I_i, L_i + \Delta L_i) \cdot \prod_{\{j|Y_i^j=0\}} (1 - F_{\omega^s}^j(I_i, L_i + \Delta L_i))) \tag{3}$$

where $F_{\omega^s}^j(I_i, L_i + \Delta L_i)$ is the $j$th element of the network output, given image $I_i$ and segmentation reference $L_i + \Delta L_i$ as input.

To optimize the model parameters and latent variables, the maximization of probability $P(\omega^l, \omega^s, \widetilde{L}|Y, I)$ equals to minimizing the cost as,

$$\mathbb{J}(\omega^l, \omega^s, \widetilde{L}) = -\frac{1}{N} \log P(\omega^l, \omega^s, \widetilde{L}|Y, I) \tag{4}$$

$$\propto \frac{1}{N} \sum_{i=1}^{N} [ ||F_{\omega^l}(I_i) - L_i - \Delta L_i||_2^2 \tag{5}$$

$$- \sum_{j} (Y_i^j \log F_{\omega^s}^j(I_i, L_i + \Delta L_i) + (1 - Y_i^j) \log(1 - F_{\omega^s}^j(I_i, L_i + \Delta L_i))) ], \tag{6}$$

where the first term (5) represents the cost for localization network training and the second term (6) is the cost for segmentation network training.

## 4.2 Iterative Joint Optimization

We propose an EM-type algorithm to optimize the learning cost $\mathbb{J}(\omega^l, \omega^s, \widetilde{L})$. As Fig. 3 illustrates, it includes two iterative steps: (i) fixing the model parameters, apply MCMC based sampling to estimate the latent variables which indicate the segmentation references $\widetilde{L}$; (ii) given the segmentation references, compute the model parameters of two networks jointly via back propagation. We explain these two steps as following.

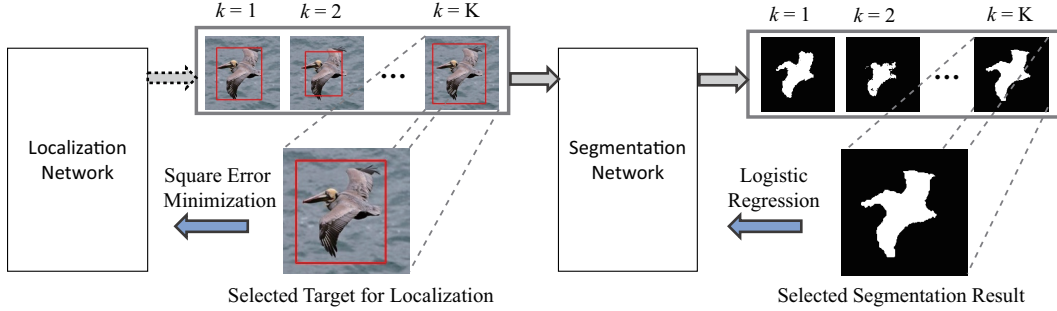

Figure 3: The Em-type learning algorithm with two steps:(i) K moves of MCMC sampling (gray arrows), the latent variables $\Delta L_i$ is sampled with considering both the localization costs (indicated by the dashed gray arrow) and segmentation costs. (ii) Given the segmentation reference and result after K moves of sampling, we apply back propagation (blue arrows) to estimate parameters of both networks.

**(i) Latent variables estimation.** Given a training image $I_i$ and current model parameters, we estimate the latent variables $\Delta L_i$. As there is no groundtruth labels for latent variables, it is intractable to estimate the distribution of them. It is also time-consuming by enumerating $\Delta L_i$ for evaluation given the large searching space. Thus we propose a MCMC Metropolis-Hastings method [24] for latent variables sampling, which is processed in $K$ moves. In each step, a new latent variable is sampled from the proposal distribution and it is accepted with an acceptance rate. For fast and effective searching, we design the proposal distribution with a data driven term based on the fact that the segmentation boundaries are often aligned with the boundaries of superpixels [1] generated from over-segmentation.

We first initialize the latent variable as $\Delta L_i = 0$. To find a better latent variable $\Delta L_i'$ and achieve a reversible transition, we define the acceptance rate of the transition from $\Delta L_i$ to $\Delta L_i'$ as,

$$\alpha(\Delta L_i \rightarrow \Delta L_i') = \min(1, \frac{\pi(\Delta L_i') \cdot q(\Delta L_i' \rightarrow \Delta L_i)}{\pi(\Delta L_i) \cdot q(\Delta L_i \rightarrow \Delta L_i')}), \tag{7}$$

where $\pi(\Delta L_i)$ is the invariant distribution and $q(\Delta L_i \rightarrow \Delta L_i')$ is the proposal distribution.

By replacing the dataset with a single sample in Eq. (1), we define the invariant distribution as $\pi(\Delta L_i) = P(\omega^l, \omega^s, \widetilde{L}_i | Y_i, I_i)$, which can be decomposed into two probabilities: $P(\omega^l, \widetilde{L}_i | Y_i, I_i)$ constrains the segmentation reference to be close to the output of the localization network; $P(\omega^s, \widetilde{L}_i | Y_i, I_i)$ encourages a segmentation reference contributing to a better segmentation mask. To calculate these probabilities, we need to perform forward propagations in both networks.

The proposal distribution is defined as a combination of a gaussian distribution and a data-driven term as,

$$q(\Delta L_i \rightarrow \Delta L_i') = \mathcal{N}(\Delta L_i' | \mu_i, \Sigma_i) \cdot P_c(\Delta L_i' | Y_i, I_i), \tag{8}$$

where $\mu_i$ and $\Sigma_i$ is the mean vector and covariance matrix for the optimal $\Delta L_i'$ in the previous iterations. It is based on the observation that the current optimal $\Delta L_i'$ has high possibility for being selected before. For the data driven term $P_c(\Delta L_i' | Y_i, I_i)$, it is computed depending on the given

image $I_i$. After over-segmenting $I_i$ into superpixels, we define $v_j = 1$ if the $j$th image pixel is on the boundary of a superpixel and $v_j = 0$ if it is inside a superpixel. We then sample $c$ pixels along the segmentation reference $\widetilde{L}'_i = L_i + \Delta L'_i$ in equal distance, then the data driven term is represented as $P_c(\Delta L'|Y, I) = \frac{1}{c}\sum_{j=1}^{c} v_j$. Thus we encourage to avoid cutting through the possible foreground superpixels with the bounding box edges, which leads to more plausible proposals. We set $c = 200$ in our experiment, and we only need to perform over-segmentation for superpixels once as pre-processing for training.

**(ii) Model parameters estimation.** As it is shown in Fig. 3, given the optimal latent variable $\Delta L$ after $K$ moves of sampling, we can obtain the corresponding segmentation references $\widetilde{L}$ and the segmentation results. Then the parameters for segmentation network $\omega^s$ is optimized via back propagation with logistic regression(as the second term (6) for Eq. (4)), and the parameters for localization network $\omega^l$ is tuned by minimizing the square error between the segmentation references and the localization output(as the first term (5) for Eq. (4)).

During back propagation, we apply the stochastic gradient descent to update the model parameters. For the segmentation network, we use an equal learning rate for all layers as $\epsilon_1$. For localization, we first pre-train the network discriminatively for classifying 1000 object categories in the Imagenet dataset [12]. With the pre-training, we can borrow the information learned from a large dataset to improve our performance. We maintain the parameters of the convolutional layers and reset the parameters of full connection layer randomly as initialization. The learning rate is set as $\epsilon_2$ for the full connection layers and $\epsilon_2/100$ for the convolutional layers.

## 5 Experiment

We validate our approach on the Saliency dataset [9, 8] and a more challenging dataset newly collected by us, namely Object Extraction(OE) dataset[1]. We compare our approach with state-of-the-art methods and empirical analyses are also presented in the experiment.

The Saliency dataset is a combination of THUR15000 [8] and THUS10000 [9] datasets, which includes 16233 images with pixelwise groundtruth masks. Most of the images contain one salient object, and we do not utilize the category information in training and testing. We randomly split the dataset into 14233 images for training and 2000 images for testing. The OE dataset collected by us is more comprehensive, including 10183 images with groundtruth masks. We select the images from the PASCAL [15], iCoseg [3], Internet [28] datasets as well as other data (most of them are about people and clothes) from the web. Compared to the traditional segmentation dataset, the salient objects in the OE dataset are more variant in appearance and shape(or pose) and they often appear in complicated scene with background clutters. For the evaluation in the OE dataset, 8230 samples are randomly selected for training and the remaining 1953 ones are applied in testing.

*Experiment Settings.* During training, the domain of each element in the 4-dimension latent variable vector $\Delta L_i$ is set to $[-10, -5, 0, 5, 10]$, thus there are $5^4 = 625$ possible proposals for each $\Delta L_i$. We set the number of MCMC sampling moves as $K = 20$ during searching. The learning rate is $\epsilon_1 = 1.0 \times 10^{-6}$ for the segmentation network and $\epsilon_2 = 1.0 \times 10^{-8}$ for the localization network. For testing, as each pixelwise output of our method is well discriminated to the number around 1 or 0, we simply classify it as foreground or background by setting a threshold 0.5. The experiments are performed on a desktop with an Intel I7 3.7GHz CPU, 16GB RAM and GTX TITAN GPU.

### 5.1 Results and Comparisons

We now quantitatively evaluate the performance of our method. For evaluation metric, we adopt the Precision, P(the average number of pixels which are correctly labeled in both foreground and background) and Jaccard similarity, J(the average intersection-over-union score: $\frac{S \cap G}{S \cup G}$, where $S$ is the foreground pixels obtained via our algorithm and $G$ is the groundtruth foreground pixels). We then compare the results of our approach with machine learning based methods such as figure-ground segmentation [22], CPMC [6] and Object Proposals [13]. As CPMC and Object Proposals generates multiple ranked segments intended to cover objects, we follow the process applied in [22] to evaluate its result. We use the union of the top $\mathcal{K}$ ranked segments as salient object prediction.

|   | Ours(full) | Ours(sim) | FgSeg [22] | CPMC [6] | ObjProp [13] | HS [32] | GC [10] | RC [9] | HC [9] |
|---|---|---|---|---|---|---|---|---|---|
| P | **97.81** | 96.62 | 91.92 | 83.64 | 72.60 | 89.99 | 89.23 | 90.16 | 89.24 |
| J | **87.02** | 81.10 | 70.85 | 56.14 | 54.12 | 64.72 | 58.30 | 63.69 | 58.42 |

Table 1: The evaluation in Saliency dataset with Precision(P) and Jaccard similarity(J). Ours(full) indicates our joint learning method and Ours(sim) means learning two networks separately.

|   | Ours(full) | Ours(sim) | FgSeg [22] | CPMC [6] | ObjProp [13] | HS [32] | GC [10] | RC [9] | HC [9] |
|---|---|---|---|---|---|---|---|---|---|
| P | **93.12** | 91.25 | 90.42 | 76.33 | 72.14 | 87.42 | 85.53 | 86.25 | 83.37 |
| J | **77.69** | 71.50 | 70.93 | 53.76 | 54.70 | 62.83 | 54.83 | 59.34 | 50.61 |

Table 2: The evaluation in OE dataset with Precision(P) and Jaccard similarity(J). Ours(full) indicates our joint learning method and Ours(sim) means learning two networks separately.

We evaluate the performance of all $\mathcal{K} \in \{1, ..., 100\}$ and report the best result for each sample in our experiment. Besides machine learning based methods, we also report the results of salient region detection methods [10, 32, 9]. Note that there are two approaches mentioned in [9] utilizing histogram based contrast(HC) and region based contrast(RC). Given the salient maps from these methods, an iterative GrabCut proposed in [9] is utilized to generate binary segmentation results.

*Saliency dataset.* We report the experiment result in this dataset as Table. 1. Our result with joint task learning (namely as Ours(full)) reaches $97.81\%$ in Precision(P) and $87.02\%$ in Jaccard similarity(J). Compared to the figure-ground segmentation method [22], we have $5.89\%$ improvements in P and $16.17\%$ in J. For the saliency region detection methods, the best results are P:$89.99\%$ and J:$64.72\%$ in [32]. Our method demonstrates superior performances compared to these approaches.

*OE dataset.* The evaluation of our method in OE dataset is shown in Table. 2. By jointly learning localization and segmentation networks, our approach with $93.12\%$ in P and $77.69\%$ in J achieves the highest performances compared to the state-of-the-art methods.

One spotlight of our work is its high efficiency in testing. As Table. 3 illustrates, the average time for object extraction from an image with our method is $0.014$ seconds, while figure-ground segmentation [22] requires $94.3$ seconds, CPMC [6] requires $59.6$ seconds and Object Proposal [13] requires $37.4$ seconds. For most of the saliency region detection methods, the runtime are dominated by the iterative GrabCut process, thus we apply its time as the average testing time for the saliency region detection methods, which is $0.711$ seconds. As a result, our approach is $50 \sim 6000$ times faster than the state-of-the-art methods.

During training, it requires around 20 hours for convergence in the Saliency dataset and 13 hours for the OE dataset. For latent variable sampling, we also try to enumerate the 625 possible proposals exhaustively for each image. It achieves similar accuracy as our approach while costs about 30 times of runtime in each iteration of training.

## 5.2 Empirical Analysis

For further evaluation, we conduct two following empirical analyses to illustrate the effectiveness of our method.

(I) To clarify the significance of joint learning instead of learning two networks separately, we discard the latent variables sampling and set all $\Delta L_i = 0$ during training, namely as Ours(sim). We illustrate the training cost $\mathbb{J}(\omega^l, \omega^s, \widetilde{L})$ (Eq. (4)) for these two methods as Fig. 4. We plot the average loss over all training samples though the training iterations, and it is shown that our joint learning

|   | Ours(full) | FgSeg [22] | CPMC [6] | ObjProp [13] | Saliency methods |
|---|---|---|---|---|---|
| Time | **0.014s** | 94.3s | 59.6s | 37.4s | 0.711s |

Table 3: Testing time for each image. The Saliency methods indicates the saliency region detection methods [32, 10, 9].

|            | Car |  | Horse |  | Airplane |  |
|------------|-------|-------|-------|-------|-------|-------|
|            | P | J | P | J | P | J |
| Ours(full) | **87.95** | **68.86** | 88.11 | 53.80 | **92.12** | **60.10** |
| Chen et al. [7] | 87.09 | 64.67 | **89.00** | **57.58** | 90.24 | 59.97 |
| Rubinstein et al. [28] | 83.38 | 63.36 | 83.69 | 53.89 | 86.14 | 55.62 |

Table 4: We compare our method with two object discovery and segmentation methods in the Internet dataset. We train our model with other data besides the ones in the Internet dataset.

method can achieve lower costs than the one without latent variable adjustment. We also compare these two methods with Precision and Jaccard similarity in both datasets. As Table. 1 illustrates, there are $1.19\%$ and $5.92\%$ improvements in P and J when we learn two networks jointly in the Saliency dataset. For the OE dataset, the joint learning performs $1.87\%$ higher in P and $6.19\%$ higher in J than learning two networks separately, as shown in Table. 2.

(II) We demonstrate that our method can be well generalized across different datasets. Given the OE dataset, we train our model with all the data except for the ones collected from Internet dataset [28]. Then the newly trained model is applied for testing on the Internet dataset. We compare the performance of this deep model with two object discovery and co-segmentation methods [28, 7] in the Internet dataset. As Table. 4 illustrates, our method achieves higher performance in the class of Car and Airplane, and a comparable result in the class of Horse. Thus our model can be well generalized to handle other datasets which are not applied in training and achieve state-of-the-art performances. It is also worth to mention that it requires a few seconds for testing via the co-segmentation methods [28, 7], which is much slower than our approach with $0.014$ seconds per image.

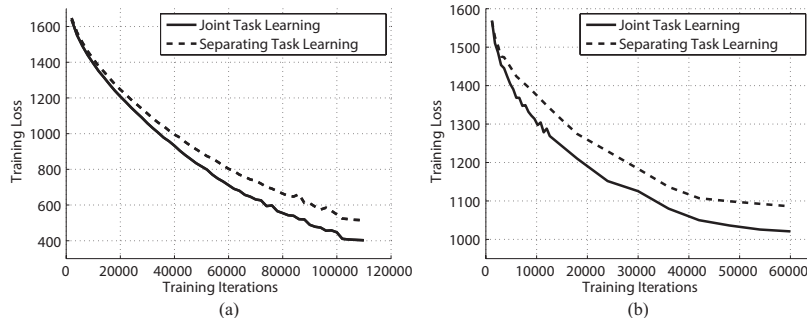

Figure 4: The training cost across iterations. The cost is evaluated over all the training samples in each dataset:(a) Saliency dataset;(b) OE dataset.

## 6 Conclusion

This paper studies joint task learning via deep neural networks for generic object extraction, in which two networks work collaboratively to boost performance. Our joint deep model has been shown to handle well realistic data from the internet. More importantly, the approach for extracting object segmentation mask in the image is very efficient and the speed is 1000 times faster than competing state-of-the-art methods. The proposed framework can be extended to handle other joint tasks in similar ways.

## Footnotes

[1]http://vision.sysu.edu.cn/projects/deep-joint-task-learning/

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
