[Reviews · NeurIPS 2014]

Submitted by Assigned_Reviewer_27

Summary:
This paper proposes a model that jointly trains two networks. One network tries
to do localization. The second network does segmentation. The first network
outputs the coordinates of a bounding box. The image is cropped using these
coordinates and resized to a fixed square size. The resulting image is given as input
to the second network. The second network predicts a pixel-wise binary mask to
segment out the foreground. The model is supervised i.e., pixel-wise segmentation
labels are given. The authors argue that if the two tasks are decoupled and a
tight bounding box around the object is used as the target for the first
network, then the predicted bounding boxes may be too tight and not provide
enough context to do segmentation. They may also cut across some region of the
foreground.

To prevent this, the authors introduce latent variables which are perturbations
in the coordinates of the bounding box. The localization net now has to predict
the perturbed bounding box. However the optimal perturbation is not known.
Therefore the model optimizes for the perturbation using a Metropolis-Hastings sampling strategy that uses super-pixels to figure out the goodness of any perturbation.

Strengths
- Shows improvements over decoupling the tasks and using no perturbation.
- The model is very efficient.
- The performance is compared to many other models. These models include ones based on hand-crafted features alone as well as ones that involve some form of learning. The model is convincingly better than these other models.

Weaknesses
- A simple baseline would be to just add a 5 or 10 pixel boundary around the predictions of the localization net. The authors do not compare to this baseline.
- A slightly stronger baseline would be to modify the predicted bounding boxes using super-pixels as a post-processing step (maybe using the same idea that is used in the data-dependent term of the MH proposal distribution). This baseline is not used either.
- In the absence of strong baselines, it is not clear what the joint optimization buys us.
- The proposal distribution's data-dependent term uses super-pixel cues in a very ad-hoc way. The term is designed to avoid cutting through super-pixel regions. However, unless the super-pixel regions are aligned with the axes of the bounding box rectangle, it seems that it would be hard to avoid cutting through super-pixels. Moreover, there may be a lot of background region going across the edges of the bounding box. It is perfectly reasonable for the bounding box to cut through super-pixels in such regions.

Quality:
The paper does not compare to simpler baseline models (as mentioned in the weaknesses section) which makes it hard to assess it contribution.

Clarity:
There seem to be many typos and grammatical errors, some of which are mentioned here.
Minor typos -
- Eq(1) The equality should probably be a proportionality.
- Eq(2) There seems to a square missing over the L2 norm.
- '... may lead to unsatisfied pixel-wise segmentation ..' unsatisfied -> unsatisfactory ?

Originality:
The idea of using Metropolis-Hastings to combine two tasks seems interesting. However, this particular application is not very compelling.

Significance:
The main idea is interesting but more experiments with stronger baselines are needed to assess the potential impact of this model.
Summary: The model is interesting and potentially useful. However, it must be compared with simpler baselines.

Submitted by Assigned_Reviewer_41

This paper offers a method for class-invariant object segmentation. The novelty of the approach lies in the model -- it is a deep neural net (DNN). It consists of two subnets. The first one performs localization up to a bounding box. Using this box, the image is cropped and a second net is used to segment the object contained within the box.

The difficulty of the above architecture lies in the way how the bounding box net and the segmentation net are connected -- the segmentation net needs an image crop dependent on the output of the bbox net. The second contribution of the paper is the use of sampling to estimate the latent bounding box crop of the segmentation net. The presented learning procedure shows a su

The authors apply the above approach for segmentation of objects -- the object type is not estimated, but the foreground mask only. It achieves state of art results against other approaches where the gap is pretty large (9%+ absolute).

All in all it is a well written paper with clear contributions. It is a good example of using deep learning for segmentation showing state-of-art performance.

Nevertheless, I have the following remarks, which must not overshadow the general positive comments above:
1. I am not sure that the Eq. (1) is the correct training objective? One does maximize the probability of the model parameters given the data, but more the probability of observing the data?!
2. The authors talk about saliency, however, in traditional sense saliency is defined as finding points of fixation. It seems more appropriate to talk about object segmentation.
3. What happens if there are more then one "salient" objects in an image? Can the approach be extended to handle multiple objects (or bounding boxes + segmentation masks)?
Summary: A novel use of deep learning for object segmentation which concatenates a localizer and a segmentation neural net and trains them jointly. Hence, I would recommend acceptance.

Submitted by Assigned_Reviewer_45

This paper presents a method to localize and segment foreground objects in images. Two convolutional networks are employed: one that predicts an object bounding box, and a second that predicts a binary segmentation mask of fixed dimension given a cropped region. Furthermore, the two networks are linked at training time via latent variables that adjust the bounding boxes given by the ground truth segmentations. Thus, the bound-box network targets are adjusted to result in better segmentation predictions by the downstream network. The latent variables are inferred using MCMC sampling, optimized alternatingly with the networks' parameters. Very good results are reported on the Saliency dataset (THUR15000+THUS10000) and a second dataset that aggregates several existing smaller segmentation datasets.

* Very good results for segmentation against fairly strong baselines
* Joint optimization with bbox adjustment moves beyond independently trained networks (and even these still perform well)

Comments/questions:

* It seems a drawback of the approach may be that it relies on predicting only a single segmentation of a salient object, making it potentially ill-suited for tasks that allow multiple guesses (e.g. imagenet), at least in its current form. How well does this handle multiple objects and clutter, and are there ways the approach may be extended to better allow for these or perform multi-object detection tasks?

* Did the independently trained networks use bbox targets that were tight around the segmentation region, or were the bboxes enlarged? Including a bit of extra context, e.g. enlarging by around 1.2x, could make these perform better.

* Similarly, I'm curious what is the mean Delta L that the algorithm finds?

* When cropping regions for the segmentation network, is the aspect ratio preserved or is the bbox reshaped to the input size?

* Fig. 1 seems a little extreme, but perhaps these are illustrative cases rather than typical ones? The segmentation network looks overly sensitive to the input region here.

* Instead of MH sampling or exhaustive enumeration, have you also tried just drawing K samples from N(Delta L') and choosing the argmax pi(Delta L')?
Summary: This is a clearly explained paper that links segmentation with bounding box localization a very nicely.
Author Feedback
Author rebuttal: First, the authors would like to thank for the area chair and three reviewers for reviewing.

1.To reviewer_27

Comment 1: The paper does not compare to simpler baseline models, which makes it hard to assess it contribution.
Response 1: Integrating two CNNs with Metropolis-Hastings is a general and principled solution for the joint task learning. As we explained at the beginning, the desired segmentation reference is unknown. Either adding a few pixels around the predicted localization (the mentioned baseline#1) or modifying the predicted localization by superpixels (the mentioned baseline#2) is an engineered and ad-hoc way, as they may cause parameter tuning on some specific data. In particular, for the baseline#1, we can use it as the initialization of our algorithm, so that we can surely obtain a better result by MCMC sampling. For the baseline#2, given the extracted superpixels, there is not a deterministic method to obtain the adjustment (e.g. enlarging or shrinking) to the prediction, and an exhaustion would be inevitable. The advantage of our model over the two simpler baselines is obvious, and we thus didn’t consider the extra empirical comparison necessary for this part. In fact, we have shown very good results against fairly strong baselines, as the Reviewer_45 commented.

Comment 2: The proposal distribution's data-dependent term uses super-pixel cues in a very ad-hoc way. …it seems that it would be hard to avoid cutting through super-pixels. …It is perfectly reasonable for the bounding box to cut through super-pixels in such regions.
Response 2: This data-dependent term is a part of the proposal distribution driving the MCMC transition, and it is designed for just accelerating the sampling. This term cannot absolutely avoid the bounding box cutting through superpixels, but it encourages the bounding box to include the complete superpixles. As the foreground superpixels often appear denser and compacter than those from background, this term tends to preserve the boundaries of foreground objects, while allows the bounding box to cut through background superpixels. Moreover, this term does not deterministically generate the bounding boxes, but proposes a probability from the image. Based on the Metropolis-Hastings method, all possibilities of placing the bounding box will be reachable during the sampling. With or without this term, it is a matter of convergence efficiency in the learning. Some MCMC-based image segmentation approaches [Tu and Zhu, IEEE TPAMI, 2002] also applied superpixels in the proposal distribution.

Comment 3: Some minor typos should be corrected.
Response 3: Thanks a lot for the remind. We will carefully polish the paper.

2.To reviewer_41

Comment 1: I am not sure that the Eq.(1) is the correct training objective? ...
Response 1: Here the CNNs’ parameters are treated as the variables to be optimized, so we formulate the objective by maximizing the posteriori probability. Also, we can estimate the parameters by maximizing likelihood estimation. Without imposing extra priors, these two forms are equally solved. Thanks for the suggestion.

Comment 2: The authors talk about saliency, ... It seems more appropriate to talk about object segmentation.
Response 2: Yes, we address the object segmentation. In recent literature [8], they define the saliency detection as identifying dominant objects from background. And one dataset we used in the experiments was originally proposed for saliency segmentation. Nevertheless, we would like to revise accordingly to highlight the object segmentation.

Comment 3: What happens if there are more than one "salient" objects in an image? Can the approach be extended to handle multiple objects...?
Response 3: There are actually a few such cases in our datasets. If these objects are close to each other, we can segment them as a whole. If they are far away from each other, we can only extract one of them out with the current implementation. It is straightforward to extend our approach to handle multiple objects: modify the localization network for multiple bounding boxes as ouputs, and apply the segmentation network to each box.

3.To reviewer_45

Comment 1: It seems a drawback of the approach may be that it relies on predicting only a single segmentation of a salient object…
Response 1: Please refer the response 3 for reviewer_41.

Comment 2: Did the independently trained networks use bbox targets that were tight around the segmentation region, or were the bboxes enlarged?...
Response 2: The bbox targets were tight around the segmentation region. We can try to enlarge the bbox as initialization for learning in the future.

Comment 3: Similarly, I'm curious what is the mean Delta L that the algorithm finds?
Response 3: For different training examples, the bounding boxes are located diversely, and the mean Delta L could be statistically meaningless. But we can add an extra experiment to visualize the distribution of Delta L over all examples.

Comment 4: When cropping regions for the segmentation network, is the aspect ratio preserved or is the bbox reshaped to the input size?
Response 4: We reshape the bbox to the input size directly, i.e. 55*55.

Comment 5: Fig.1 seems a little extreme, but perhaps these are illustrative cases rather than typical ones?...
Response 5: The cases in Fig. 1 are selected for illustration, but they are indeed generated by our approach. In fact, the input reference is critical to segmentation, just like some interactive segmentation methods.

Comment 6: Instead of MH sampling or exhaustive enumeration, have you also tried just drawing K samples from N(Delta L')...?
Response 6: The model learning takes quite a while, e.g. >20 hours, so we didn’t conduct this implementation. We believed the data-driven MH method with would be a more general solution. However, it is worth implementing more experiments to verify the latent variable sampling.